# Genome-Wide Identification of Gramineae Brassinosteroid-Related Genes and Their Roles in Plant Architecture and Salt Stress Adaptation

**DOI:** 10.3390/ijms23105551

**Published:** 2022-05-16

**Authors:** Jinyong Huang, Shengjie Ma, Kaiyan Zhang, Xiaohan Liu, Linlin Hu, Wenming Wang, Liwei Zheng

**Affiliations:** School of Agricultural Sciences, Zhengzhou University, Zhengzhou 450001, China; jinyhuang@zzu.edu.cn (J.H.); 053186255812ma@163.com (S.M.); zkyan678@163.com (K.Z.); lxh12310849@163.com (X.L.); junfu@gs.zzu.edu.cn (L.H.); zxc807037@163.com (W.W.)

**Keywords:** brassinosteroid, plant architecture, salt, Gramineae, gene family

## Abstract

Brassinosteroid-related genes are involved in regulating plant growth and stress responses. However, systematic analysis is limited to Gramineae species, and their roles in plant architecture and salt stress remain unclear. In this study, we identified brassinosteroid-related genes in wheat, barley, maize, and sorghum and investigated their evolutionary relationships, conserved domains, transmembrane topologies, promoter sequences, syntenic relationships, and gene/protein structures. Gene and genome duplications led to considerable differences in gene numbers. Specific domains were revealed in several genes (i.e., *HvSPY*, *HvSMOS1*, and *ZmLIC*), indicating diverse functions. Protein-protein interactions suggested their synergistic functions. Their expression profiles were investigated in wheat and maize, which indicated involvement in adaptation to stress and regulation of plant architecture. Several candidate genes for plant architecture (*ZmBZR1* and *TaGSK1/2/3/4-3D*) and salinity resistance (*TaMADS22/47/55-4B*, *TaGRAS19-4B*, and *TaBRD1-2A.1*) were identified. This study is the first to comprehensively investigate brassinosteroid-related plant architecture genes in four Gramineae species and should help elucidate the biological roles of brassinosteroid-related genes in crops.

## 1. Introduction

Leaf angle (LA), the angle between the leaf blade and stem, and plant height (PH) are important agronomic factors in determining plant architecture [1]. LA shapes plant architecture and influences light interception, which contribute to increasing crop yield and productivity [2]. PH is associated with grain yield and is involved in crop lodging resistance [3]. Soil salinization inhibits crop growth and development, which, in turn, reduces yield production. With the rapid increase in the global population, ever-increasing food demand has become a considerable challenge. Therefore, uncovering the mechanism regulating plant architecture and salinity resistance is critical.

Brassinosteroid (BR) phytohormones play various roles in plant growth and stress adaptation [4]. Abundant functional information on BRs has been obtained for *Arabidopsis* and rice [5,6,7], suggesting roles in regulating plant architecture and increasing LA and PH. Notably, both LA and PH are influenced by BR biosynthesis, signaling, and downstream-related genes.

Several BR biosynthesis genes (*OsDWARF4*, *OsDWARF4L1*/*D11*, *OsDWARF2* (*OsD2*), *OsD3*, and *OsDWARF*/*OsBRD1*) control endogenous BR content and affect erect leaf phenotype and stem elongation [1,2,8,9,10,11]. In rice, the BR signaling receptors BRASSINOSTEROID INSENSITIVE 1 (BRI1) and BRI1-ASSOCIATED KINASE 1 (BAK1) participate in lamina joint bending and PH [12,13]. Furthermore, the BR signaling negative regulator *glycogen synthase kinase1/2/3/4* (*OsGSK1/2/3/4*) indirectly regulates lamina inclination [14]. BRASSINAZOLE-RESISTANT 1 (BZR1), a key BR downstream transcription factor, participates in the regulation of BR-responsive genes, PH, and erect leaves [15]. BR downstream genes [Rice *SPINDLY* (*OsSPY*), *GA-stimulated transcript* (*OsGSR1*), ELT1, *SMALL ORGAN SIZE 1* (*SMOS1*), *DLT*, Ovate family protein 1 (*OFP1)*, *OFP8*, *LEAF and TILLER ANGLE INCREASED CONTROLLER* (*LIC*), *U-type cyclin* gene (*Os**CYC U4;1*), *TUD1*, *BRASSINOSTEROID UPREGULATED1* (*BU1*), *BU1-LIKE1* (*BUL1*), *INCREASED LEAF INCLINATION 1* (*ILI1*), *IBH1*, *GA INSENSITIVE REPRESSOR OF GAI and SCARECROW 19* (*OsGRAS19*), *MADS22*, *MADS47*, *MADS55*, *XIAO*, *LC2*, and auxin-response factors (*ARF11* and *ARF19*)] also play significant roles in shaping plant architecture. For example, *OsSPY* and *OsGSR1* are positive and negative regulators of BR biosynthesis and GA signaling, respectively, and both influence LA, PH, and grain size [16,17]; *ELT1*, a receptor-like protein, inhibits BR signal receptor endocytosis, and its overexpression in rice enhances BR response and LA [18].

The above BR-related genes also play pivotal roles in environmental stress responses. For example, GSKs and BES1/BZR1 have emerged as key factors antagonizing drought, cold, and heat responses [19]. ABA plays an important role in regulating responses to abiotic stress, and BR and ABA exhibit an antagonistic relationship mediated by *BRI1*, *BAK1*, and *SnRK2s* [20]. Furthermore, overexpression of *AtDWARF4* in *Brassica napus* enhances drought tolerance [21].

While the above BR-related genes are known regulators of LA, PH, and stress responses in rice, their functions in plant architecture and salt stress tolerance in *Triticum aestivum*, *Zea mays*, *Hordeum vulgare*, and *Sorghum bicolor* remain unclear. In the current study, we conducted a genome-wide identification of BR-related genes in these species and determined chromosomal location, phylogenetic analysis, sequence alignment, gene/protein structure, transmembrane structure, synteny relationship, Ka/Ks values, promoter sequences, and protein-protein interactions. In addition, we investigated their potential involvement in regulating plant architecture and salt tolerance in wheat and maize using transcriptome data.

## 2. Results

### 2.1. Identification of BR-Related Genes in Gramineae

We identified 68, 34, 33, and 29 BR-related genes (DWARF4/D11, D2/3, SPY, GSR1, BRD1, ELT1, BRI1, BAK1, GSK1/2/3/4, BZR1, SMOS1, DLT, OFP1/8, LIC, CYC U4;1, TUD1, D1, BU1/ILI1/BUL1, IBH1, GRAS19, MADS22/47/55, XIAO, LC2, and ARF11/19) in T. aestivum, Z. mays, H. vulgare, and S. bicolor (Figure 1a). Twice as many BR-related genes were identified in wheat than in rice. The genes were named according to their chromosomal positions (Appendix A). Detailed information regarding the Gramineae BR-related genes is listed in Appendix A. Based on their index values, most Gramineae proteins were likely unstable (Figure 1b), except for OsSPY, TaSPY-6A, HvSPY, ZmSPY, SbSPY, HvGSR1, HvBRD1-1, HvBRD1-2, OsBRI1, TaBRI1-3A, TaBRI1-3B, TaBRI1-3D, HvBRI1, ZmBRI1-1, ZmBRI1-2, SbBRI1, TaBAK1-7B, HvBAK1, and HvD1 (Appendix A). 

### 2.2. Multiple Sequence Alignment, Secondary Structure, and Phylogenetic Analyses of BR-Related Proteins

Conserved domains were identified from multiple protein sequence alignments (Appendix A). For example, anchor regions and proline-rich domains were found in the N terminal of DWARF4/D11, while A-C and heme-binding domains were found in the C terminal (Appendix A). Membrane anchor regions and proline-rich, dioxygen-binding, steroid-binding, and heme-binding domains were found in the D2/D3 proteins (Appendix A). Furthermore, TPR, CD I, and C II domains were identified in the N to C terminals of SPY proteins (Appendix A). However, several proteins shared specific motifs with their homologous ones. For example, there was an incomplete TPR domain in HvSPY, but no CD II domain (Appendix A); the AKER and AP2-R1 domains were lost in HvSMOS1 (Appendix A); and the CCCH domain was not in ZmLIC (Appendix A).

In terms of secondary structures, the proteins were similar to their rice homologs (Appendix A). For example, there were 50% alpha-helices, 35% random coils, 11% extended strands, and 4.5% beta-turns in the DWARF4 and D11 proteins. According to phylogenetic analysis, most *T. aestivum*, *H. vulgare*, *Z. mays*, and *S. bicolor* proteins were highly homologous to the corresponding rice proteins (Figure 2). For example, all XIAOs were clustered in subgroup A; BRI1 proteins were divided into two classes in subgroup B; and SbBAK1 shared a relatively distant relationship with other genes in group C.

### 2.3. Transmembrane Structure, Synteny, Gene Structure, Promoter, and Protein-Protein Interaction Network Analyses

Transmembrane analysis showed that about half of the BR-related plant architecture proteins shared transmembrane structures (Figure 1c and Appendix A). Tandem duplications were identified in *TaBRD1s* (Figure 1d and Appendix A). In total, 37 pairs of segmentally duplicated genes were found in the four Gramineae species, including 33 in *T. aestivum*, one in *H. vulgare*, and three in *Z. mays* (Figure 1e). To analyze the evolutionary relationships and expansion patterns of the BR-related genes, we compared the four Gramineae species with rice and identified 14, 10, 15, and 19 pairs of syntenic orthologous genes, respectively (Appendix A). The Ka, Ks, and Ka/Ks values were calculated for tandem and segmental gene pairs to identify selective pressure (Appendix A). The Ka/Ks ratios of most tandem and segmental gene pairs were less than 1, indicating that they experienced negative selection during evolution.

Most genes contained introns (Figure 1f) and genes in common pairs shared similar exon-intron structures (Appendix A). BR-related *cis*-acting elements (BRRE, G-box, E-box, GATGTG, and CTCGC) were identified in the promoters of the BR-related genes (Appendix A). To identify interactions among *T. aestivum* proteins, a protein network was constructed using rice orthologs (Appendix A), which showed that most proteins interacted with others. However, several proteins, such as TaMADS22/47/55-4D, TaGSR1-7D, and TaLIC-7B, had no partner. Like the wheat BR-related plant architecture proteins, only eight maize proteins (ZmLC2, ZmOFP8, ZmGRAS19, ZmIBH1, ZmCYC U4;1, ZmMADS22/47/55-2, ZmOFP1, and ZmLIC) showed connections with other proteins (Appendix A).

### 2.4. Expression Analysis of Maize BR-Related Genes in Regulating PH

Three different maize hybrids, with extremely distinct PH [i.e., low (L), middle (M), and high (H) groups], were used to conduct RNA sequencing (RNA-seq) [22]. These plants showed increased PH from stages 1 to 3 [i.e., jointing stage (1), big flare period (2) and tasseling stage (3)]. Cluster analysis of the maize BR-related genes was performed at the three key stem developmental stages. Maize BR-related genes were grouped into five clusters. The expression of three genes in cluster 1 decreased in the maize hybrids over time, showing the opposite trend to PH (Figure 3a,b). Among these genes, *ZmBZR1* showed the highest expression level (Figure 3c). Coronatine (COR), a functional and structural analog of jasmonic acid (JA), can improve maize lodging resistance by inhibiting ninth internode elongation [23]. The meristem (M, basal 0–1 cm between internode) and elongation regions (E, basal 1–2 cm between internode) of the ninth internode were collected for RNA-seq. In the M and E regions, the expression patterns of the BR-related genes over time were identified in the control and COR-treated groups (Figure 4a,b,f,g). In the M region, nine BR-related genes (two in cluster 2, three in cluster 4, and four in cluster 5) showed a close relationship with PH and were induced or repressed by COR (Figure 4a). For example, *ZmLIC* and *ZmLC2* in cluster 2 increased over time in the control group but were initially inhibited and then induced in the COR-treated group. Details on these genes are presented in Figure 4c–e, with highly expressed genes (*ZmLC2*, *ZmXIAO*, *ZmBZR1*, and *ZmTUD1*) marked with asterisks. In the E region, genes in clusters 2 and 6 sharing close links with PH showed different expression patterns between the control and COR-treated groups. In cluster 2, *ZmBRI1-1*, *ZmGSK1/2/3/4-2*, and *ZmMADS22/47/55-1* gradually increased from E1C to E4C, whereas their expression levels became stable after COR treatment at the same time points (Figure 4f,h). The transcription patterns of the seven genes in cluster 6 decreased over time in the control group but fluctuated slightly in the COR-treated group (Figure 4f,i). In addition, 14 maize BR-related genes were either down-regulated (cluster A) or up-regulated (cluster B) at most time points in the COR-treated group (Figure 4j). Based on gene co-expression network analysis, *ZmGSK1* was identified as a likely key regulator of COR-mediated internode elongation (Figure 4k).

### 2.5. Transcriptome Analysis of Wheat BR-Related Genes in Regulating LA and PH

In wheat, BR treatment enhanced LA from days 2 to 8, especially at days 5, 7, and 8 (Figure 5a). To investigate the potential functions of wheat BR-related genes in this process, their expression patterns were analyzed. In total, 12 genes were down-regulated by BR at different time points (Figure 5b). The expression profiles of BR-related genes over time were also identified in both the control and BR-treated groups (Figure 5c,d). These genes were clustered into eight and five clusters over time in the control and BR-treated wheat seedlings, respectively. In the control group, *TaD1-1B*, *TaOFP8-3D*, *TaOFP8-3B*, and *TaARF11-2D* showed consistent variation trends with LA with time (Figure 5a,c,e). However, the expression levels of these four genes were very low (Figure 5f). In the BR-treated group, genes in cluster 5, especially *TaGSK1/2/3/4-3A* and *TaGSK1/2/3/4-3D*, may be negatively associated with BR-induced lamina inclination (Figure 5a,d,g,h).

Transcriptome analysis of wild-type (WT) and dwarf mutants (dm1, dm2, dm3, and dm4) of wheat was performed, showing that wheat PH followed the order WT > dm1 > dm2 > dm3 > dm4 [24]. Here, 12 wheat BR-related genes gradually decreased in WT, dm1, dm2, dm3, and dm4 (Figure 6a,b). Among these genes, *TaGSK1/2/3/4-3D* and *TaELT1-6D* shared relatively high expression (Figure 6c). In addition, *TaBU1/TaILI1/TaBUL1-7D* was down-regulated in WT compared to the four dwarf mutants (Figure 6d). Other differentially expressed genes (DEGs) included *TaCYC U4;1-1B*, *TaBRD1-2D.2*, *TaOFP8-3D*, *TaOFP8-3A*, and *TaARF11-2B* (Figure 6d).

### 2.6. Expression Analyses of Wheat BR-Related Genes in Response to Salt Stress

Qing Mai 6 (QM) and Chinese Spring (CS) are salt-tolerant and salt-sensitive wheat varieties, respectively. Here, QM and CS were used for salt stress treatment. Their roots were collected at 6, 12, 24, and 48 h after salt treatment for RNA-seq [25]. After salt treatment, the wheat BR-related genes in CS could be divided into four clusters over time (Figure 7a). The expression patterns of genes in cluster 4 were similar, with reduced chlorophyll content (Figure 7b,g), and seven genes (*TaGSK1/2/3/4-3A*, *TaGSK1/2/3/4-3D*, *TaARF19-7B*, *TaARF19-7A*, *TaARF19-7D*, *TaIBH1-1A*, and *TaBRD1-2A.1*) showed relatively high expression in this cluster (Figure 7c). In QM, genes in cluster 2 were highly correlated with the salt-related trait (Figure 7d,e,g), and *TaGRAS19-4B* showed relatively high expression.

After salt treatment, most wheat BR-related genes were differentially expressed in CS and QM (Figure 8a). Throughout the experiment, five genes (*TaCYC U4;1-1B*, *TaBRD1-2A.1*, *TaMADS22/47/55-4B*, *TaBRD1-2B.2*, and *TaTUD1-4D*) responded to salt in both CS and QM. Several other genes, including *TaARF19-7B*, *TaARF19-7D*, and *TaBRD1-2B.3*, were regulated by salt in either CS or QM. For example, *TaMADS22/47/55-4A* was down-regulated at all time points in the salt-treated CS seedlings, and slightly affected by salt treatment at 12 h in QM. Their expression profiles were also compared between CS and QM under control and NaCl-treated conditions, respectively (Figure 8b). We found that *TaARF11-2D* was lowly expressed in both control and salt-treated CS samples. Furthermore, *TaDWARF4-4A* expression was relatively consistent in CS and QM under control conditions but was induced in QM compared to CS after salt treatment. According to connectivity and logFC, *TaDLT-4A* and *TaMADS22/44/57-4B* were identified as core genes in the co-expression network (Figure 8c).

Weighted gene co-expression network analysis (WGCNA) is widely used to identify candidate genes based on associations between a gene set and phenotype. According to scale independence and mean connectivity, WGCNA was performed with a power value of 18 (Figure 9a). Module-trait relationship analysis revealed that salt-related traits (i.e., root length and reactive oxygen species (ROS) accumulation) were closely related to the ‘antiquewhite4’, ‘darkred’, ‘darkmagenta’, ‘cyan’, and ‘lavenderblush3’ modules (Figure 9b). We identified 11 genes (*TaBZR1-2D*, *TaD2/3-3A*, *TaBRI1-3A*, *TaBRI1-3B*, *TaD2/3-3D*, *TaGSK1/2/3/4-3D*, *TaBRI1-3D*, *TaMADS22/47/55-4B*, *TaSPY-6A*, *TaLC2-6A*, and *TaLC2-6B*) in the ‘darkred’ module, and one gene (*TaBZR1-2B*) in the ‘cyan’ module (Appendix A). Cytoscape representation of the genes showed high interactions with the BR-related genes (weight ≥ 0.3) in the ‘darkred’ module (Figure 10a and Appendix A). As shown in Figure 10a, *TaMADS22/47/55-4B* was a hub gene. The functions of genes in the ‘darkred’ module are presented in Appendix A. Two genes (*TraesCS3B03G0684600* and *TraesCS3D03G0554900*) encoding *TaAKT1*, an important channel in plants for absorbing potassium ions under salt stress [26], were highly related to *TaMADS22/47/55-4B* (Figure 10a and Appendix A). As a transcription factor, MADS-box binds at consensus recognition sequences (CArG box) [27], and we found a CArG box in the promoter of the *TaAKT1* gene (TraesCS3D03G0554900) (Figure 10b). Genes in the ‘cyan’ module are listed in Appendix A. These genes showed high connectivity with *TaBZR1-2B* (Appendix A), but no salt-related genes were identified (Appendix A).

## 3. Discussion

### 3.1. Comparison of BR-Related Genes among Gramineae Species

A total of 35 BR-related genes have been identified in rice. Here, 68 corresponding genes were identified in wheat, showing a one-fold increase compared to rice (Figure 1a). We identified two tandem and 33 segmental BR-related plant architecture gene duplications in wheat (Figure 1d,e). Gene duplications can lead to the evolution of species [28]. Moreover, the wheat genome has undergone genome-wide duplications [29]. Therefore, the expansion and evolution of BR-related genes in wheat may have been induced by gene and genome-wide duplications.

Phylogenetic and syntenic analyses of rice-wheat, rice-maize, rice-barley, and rice-sorghum were determined to predict the functions of the wheat, maize, barley, and sorghum BR-related genes based on rice orthologs (Figure 2 and Appendix A). The wheat, maize, barley, and sorghum BR-related genes in pairs might originate from common ancestors with rice BR-related genes, indicating their similar roles. In rice, BR-related genes are involved in regulating LA and PH [1,12,14,15,16,18,30,31,32]. Therefore, their homologous genes are likely to patriciate in regulating plant architecture.

Gene functions are closely associated with their domains. Therefore, typical domains of the wheat, maize, barley, and sorghum BR-related genes were compared with those of rice. Several BR-related genes, including *HvBZR1*, *HvSPY*, *HvSMOS1*, *ZmLIC*, *HvD1*, *TaMADS22/47/55-4A*, *HvMADS55*, *ZmXIAO*, and *HvARF19*, shared distinct domains (Appendix A). Therefore, these BR-related genes may vary among Gramineae species.

### 3.2. Putative Functions of Maize and Wheat BR-Related Genes in Plant Architecture and Salt Adaptation

PH is closely associated with maize yield and lodging resistance. Here, three maize hybrids with distinct PH were selected to identify PH-related candidate genes. We found that the expression levels of *ZmBZR1*, *ZmD3*, and *ZmILI1* decreased over time, in contrast to the increase in PH (Figure 3a,b), indicating that these genes may play negative roles in regulating PH. *ZmBZR1* showed the highest expression level (Figure 3c), and therefore may be a key gene. As COR can effectively reduce maize height, we studied the potential roles of the maize BR-related genes in COR-mediated internode shortening and identified their time-series expression patterns (Figure 4). Results showed that several genes, including *ZmLC2*, *ZmXIAO*, *ZmBZR1*, *ZmTUD1*, *ZmBRI1-1*, *ZmGSK1/2/3/4-2*, *ZmMADS22/47/55-1*, *ZmELT1*, and *ZmBAK1*, were likely to participate in this process. BZR1 is a positive regulator of BR signaling, and *bzr1* loss-of-function produces a dwarf mutant [33]. In rice, BZR1 directly binds to the promoters of GA biosynthesis genes to affect GA biosynthesis, cell elongation, and PH [34]. The tomato *bri1* mutant exhibits short stem and internodes [35]. Rice *bri1* is insensitive to BR and severe dwarfism [36]. *Zmbri1*-RNAi plants show short internodes and dwarf stature [37]. Down-regulating *GSK2* to release its repression on BZR1 could increase PH in rice [15]. The gain-of-function *elt1-D* mutant reduces PH and leaf inclination in rice [18]. In *Arabidopsis*, *bak1* mutation shows a semi-dwarf phenotype [38]. *OsBAK1* is also involved in BR signal transduction and plant architecture [39]. Our study indicated that these genes (especially *ZmBZR1*) are important for maize architecture, and future studies should focus on their exact roles.

In rice, lamina inclination is tightly related to BR [40]. However, whether BR affects LA in wheat remains unknown. In this study, exogenous BR markedly enhanced lamina inclination in the wheat seedlings (Figure 5a). In the control group, four BR-related genes showed a similar LA trend (Figure 5c,e). However, their expression levels were quite low (Figure 5f), and they should be carefully identified as candidates. In total, 31 BR-related genes, especially *TaGSK1/2/3/4-3A* and *TaGSK1/2/3/4-3D*, were closely connected with lamina inclination (Figure 5d,g,h). In rice, GSK2 targets many substrates, such as *CYC U4;1*, *GRF4*, *LIC*, *BZR1*, *RLA1*, *DLT*, and *OFP8*, to regulate lamina inclination [41]. Thus, previous study and our trend analysis provided valuable information for revealing the function of *TaGSK1/2/3/4-3A* and *TaGSK1/2/3/4-3D*. RNA-seq was also used to identify wheat candidate genes involved in regulating PH (Figure 6). Based on our findings, *TaGSK1/2/3/4-3D*, *TaELT1-6D*, and *TaBU1/TaILI1/TaBUL1-7D* may be important for stem elongation (Figure 6). At the protein level, GSK2 targets LIC, BZR1, and RLA-DLT complexes to affect PH in rice [42]. Furthermore, under BZR1 control, *BU1* and *ILI1* have important effects on stem growth in rice and *Arabidopsis* [43]. Information on *ELT1*-mediated PH is limited. Thus, the potential functions of *TaGSK1/2/3/4-3D*, *TaELT1-6D*, and *TaBU1/TaILI1/TaBUL1-7D* should be confirmed in future studies.

Salt strongly influences wheat yield and quality, but the relationship between BR-related genes and salt tolerance in wheat remains unknown. Here, we identified seven BR-related genes (*TaGSK1/2/3/4-3A*, *TaGSK1/2/3/4-3D*, *TaARF19-7A*, *TaARF19-7B*, *TaARF19-7D*, *TaIBH1-1A*, and *TaBRD1-2A.1*) that may regulate salt tolerance in CS according to trend analysis (Figure 7a–c,g). In QM, *TaGRAS19-4B* was identified as a likely candidate gene for salt stimuli (Figure 7d–g). DEG analysis showed that *TaCYC U4;1-1B*, *TaBRD1-2A.1*, *TaGRAS19-4B*, *TaMADS22/47/55-4B*, *TaBRD1-2B.2*, *TaTUD1-4D*, *TaARF11-2D*, and *TaDWARF4-4A* may play critical roles in wheat salt stress regulation (Figure 8). Based on WGCNA, *TaMADS22/47/55-4B* was mainly associated with salinity (Figure 9). Combining trend and DEG analyses, the functions of *TaBRD1-2A.1* and *TaGRAS19-4B* should be explored in future study. Short time-series expression analysis and WGCNA showed that *TaMADS22/47/55-4B* plays a core role in regulating wheat resistance to salt stress.

## 4. Materials and Methods

### 4.1. Identification, Chromosomal Location, Chemical Characterization, Subcellular Localization, Multiple Sequence Alignment, and Protein Structure Analyses of BA-Related Genes

Known rice protein sequences of DWARF4/D11, D2/3, SPY, GSR1, BRD1, ELT1, BRI1, BAK1, GSK1/2/3/4, BZR1, SMOS1, DLT, OFP1/8, LIC, CYC U4;1, TUD1, D1, BU1/ILI1/BUL1, IBH1, GRAS19, MADS22/47/55, XIAO, LC2, and ARF11/19 were used to BLAST the *T. aestivum* (15 May 2022, http://plants.ensembl.org/Triticum_aestivum/Info/Index), *Z. mays* (15 May 2022, https://genome.jgi.doe.gov/portal/pages/dynamicOrganismDownload.jsf?organism=ZmaysPH207), *H. vulgare* (15 May 2022, https://genome.jgi.doe.gov/portal/pages/dynamicOrganismDownload.jsf?organism=Hvulgare), and *S. bicolor* (15 May 2022, https://genome.jgi.doe.gov/portal/pages/dynamicOrganismDownload.jsf?organism=Sbicolor) protein databases. Pfam (15 May 2022, http://pfam.xfam.org/search/sequence) was used to confirm the characteristic domains. ‘Gene Location Visualize’ in TBtools [44] was used to visualize chromosomal locations. Peptide length, molecular weight, isoelectric point (pI), instability index, aliphatic index, and grand average of hydropathicity (GRAVY) were calculated using the ExPASy program (15 May 2022, http://web.expasy.org/protparam/). Subcellular locations were predicted using the Plant-mPLoc online tool (15 May 2022, http://www.csbio.sjtu.edu.cn/bioinf/plant-multi/). DNAMAN (v6.0, LynnonBiosoft, San Ramon, CA 94583, USA) was used to align multiple protein sequences to identify conserved domains in each BR-related gene. Using the NPS (15 May 2022, https://npsa-prabi.ibcp.fr/cgibin/npsa_automat.pl?page=npsa_sopma.html) and PHYRE server (15 May 2022, v.2.0, http://www.sbg.bio.ic.ac.uk/phyre2/html/page.cgi?id=index), three-dimensional protein structures were predicted.

### 4.2. Phylogeny, Transmembrane Helices, Synteny, Gene Structure, Cis-Acting Elements in Promoters, Protein-Protein Interaction Networks, and Trend and Gene Co-Expression Network Analyses

Using MEGA X (v 10.1.6, University of Pennsylvania), a phylogenetic tree was constructed with BR-related plant architecture proteins. Transmembrane helices were predicted with the TMHMM server (15 May 2022, v.2.0, http://www.cbs.dtu.dk/services/TMHMM/). Synteny and exon/intron structure analyses were completed using ‘One Step MCScanS’ and ‘Visualize Gene Structure (Basic)’ in TBtools [44]. BR-responsive elements, including BRRE (CGTGT/CG), G-box (CACGTG), E-box (CACTTG), GATGTG, and CTCGC, were identified in the promoter region (-3-kb upstream of the transcription start site). STRING (15 May 2022, v10, http://string-db.org/) (option value > 0.800) was used to construct a protein interaction network according to the interolog proteins from rice. Using Genesis (15 May 2022, v1.8.1, http://genome.tugraz.at/projects.shtml), time-series expression data were analyzed to identify the best-fit model and phase of expression. Gene expression correlations were calculated with the cor. test function in R (v4.1). Using high-quality genes, WGCNA was performed in R (WGCNA package v1.51). To reduce hierarchical clustering, the dynamic tree cut algorithm, with a coefficient of variation cut-off of 0.25, power β of 18, and minimum module size of 30 genes, was used to filter genes with low variation among the samples. Significant module-trait relationships with root length and ROS accumulation were identified by calculating the module eigengene value. Gene co-expression networks were visualized with Cytoscape (v3.8.2, https://cytoscape.org/download.html 13 April 2022). 

### 4.3. Plant Materials, BR Treatment, and Transcriptome Analysis

Using water in a 50-mL conical flask, wheat seeds (Bainong 321) were shaken until the seeds with a radicle length of approximately 5 mm. The germinated seeds were immersed in wet gauze above water for 5 d at room temperature. The wheat seedlings were then hydroponically grown with 1/4 Hoagland nutrient solution in an illuminated growth chamber using published culture conditions [45]. After approximately 2 days of growth, the second leaf of the wheat seedlings became visible, and 3 μM 24-epitestosterone (BR) was added to the nutrient solution. Lamina joints for transcriptome (RNA-seq) analysis were sampled on days 2, 3, 5, 7, and 8. Three replicates containing at least 18 plants each were used for each time point.

Total RNA was extracted from the lamina joints using TRIzol reagent (Cwbio, Beijing, China) according to the manufacturer’s instructions. The cDNA was synthesized using a reverse transcription kit (Cwbio, Beijing, China). Using Poly-A Purification TruSeq library reagents (Illumina), barcoded cDNA libraries were constructed and then sequenced on the NovaSeq 6000 platform (Illumina). For each library, ~10 Gb of high-quality 150-bp paired-end reads were generated. The raw RNA-seq data used for identifying BR-mediated LA can be found in the NCBI Sequence Read Archive database (accession number SRP157960). The fastp software (v0.20.1) was used to evaluate the overall sequencing quality of the reads and to remove low-quality reads in each sample. Hisat2 (v2.1.0) and SAMtools (v1.6) were used to align the high-quality reads to the wheat reference genome sequence IWGSC RefSeq (v2.1) [46]. Stringtie (v1.3.3b) was used to calculate the expression levels of high-confidence genes in each sample. The R package “edgeR” was used to identify DEGs between the control and BR-treated groups with the parameters “|log2 (fold change)| >= 1 and *p* < 0.05”.

## 5. Conclusions

In summary, we performed a genome-wide analysis of BR-related genes in four Gramineae species. Their chromosomal location, conserved domain, secondary structure, transmembrane topology, gene structure, promoter sequence, synteny, and protein-protein interaction networks were analyzed. The expression profiles of BR-related genes in different maize and wheat lines with highly distinct PH and responses to COR and BR were delineated (Figure 3, Figure 4, Figure 5 and Figure 6). As shown in Figure 3, Figure 4, Figure 5 and Figure 6, *ZmBZR1* and *TaGSK1/2/3/4-3D* appear to be the main regulators of maize and wheat height, respectively. Based on comprehensive analysis (Figure 7, Figure 8, Figure 9 and Figure 10), *TaBRD1-2A.1*, *TaGRAS19-4B*, and *TaMADS22/47/55-4B* may participate in wheat resistance to salt stress. Future studies should validate the functions of the above-mentioned candidate genes.

## Figures and Tables

**Figure 1 ijms-23-05551-f001:**
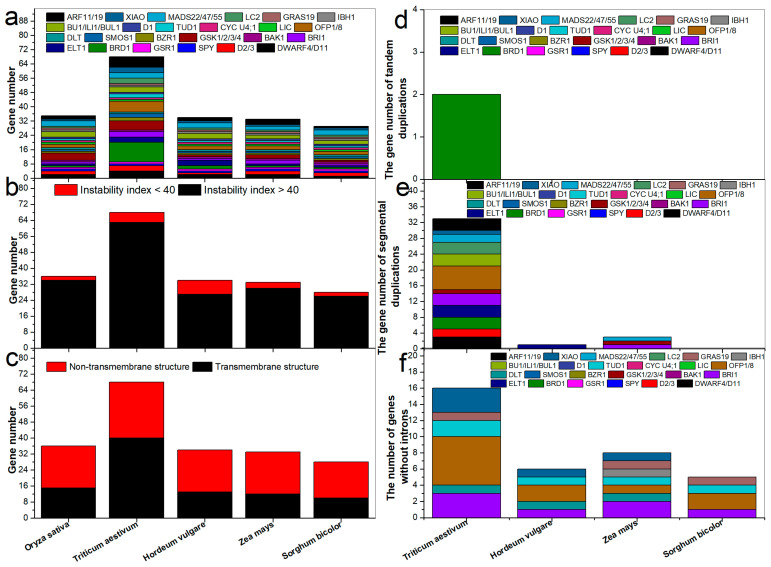
Number of BR-related genes among different species. (**a**) Gene number of each BR-related plant architecture gene in rice and four Gramineae species. (**b**) Instability index of BR-related plant architecture genes in rice and four Gramineae species. (**c**) Transmembrane structure of BR-related plant architecture genes in rice and four Gramineae species. (**d**) Gene number of tandem duplications. (**e**) Gene number of segmental duplications. (**f**) Number of genes without introns.

**Figure 2 ijms-23-05551-f002:**
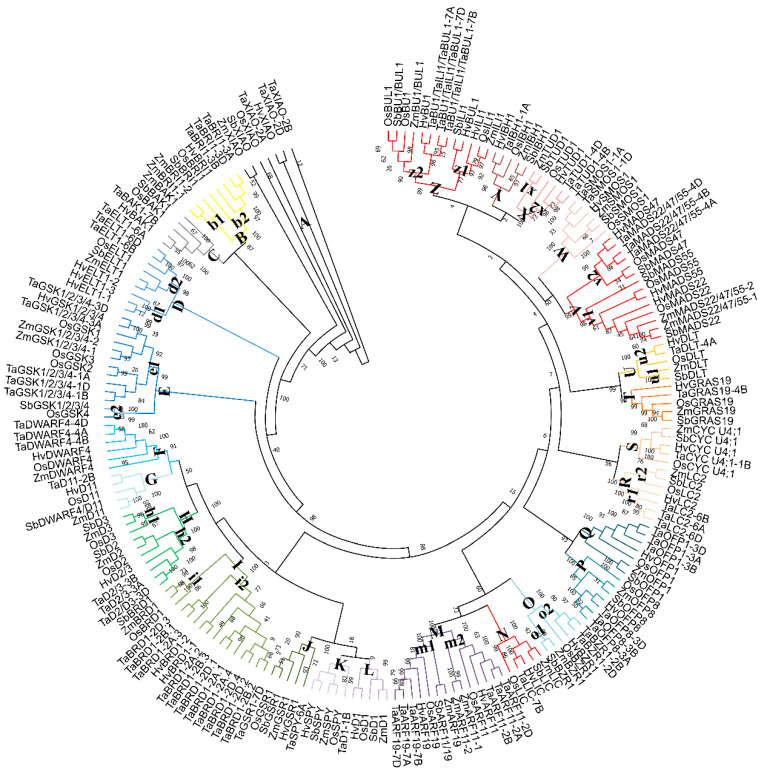
Phylogenetic tree of BR-related genes. Each Brassicaceae specie BR-related genes were represented by the bold letters.

**Figure 3 ijms-23-05551-f003:**
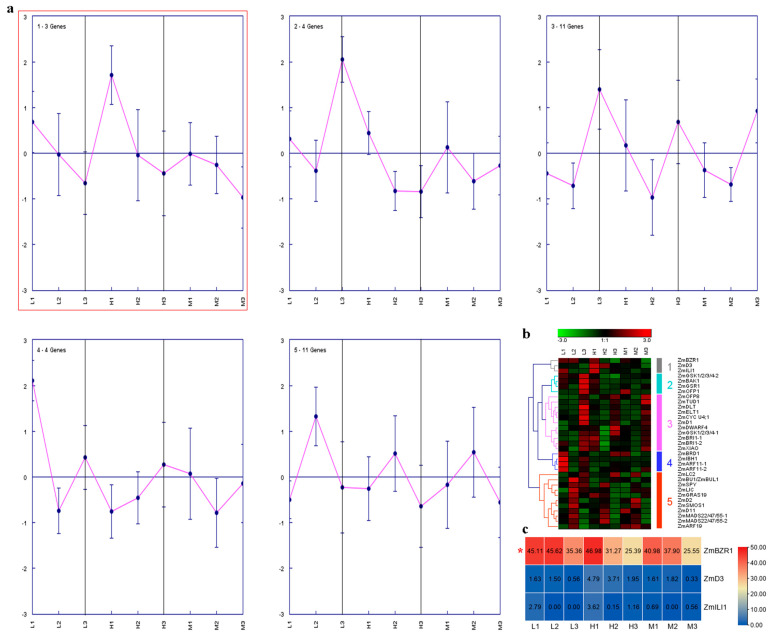
Trend analysis of BR-related genes in three different maize hybrids with distinct plant heights. (**a**) Clusters were obtained based on expression data of BR-related genes in three different maize hybrids at the jointing stage (1), big flare period (2), and tasseling stage (3). (**b**) Heatmap of all clusters. (**c**) Expression levels of other BR-related genes in three different maize hybrids at three stages. Genes with high expression are marked with a red asterisk.

**Figure 4 ijms-23-05551-f004:**
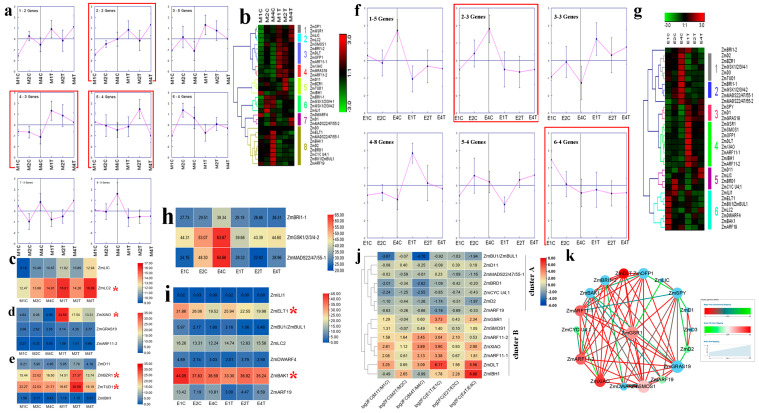
Trends and expression analyses of maize BR-related genes in response to coronatine. (**a**) Clusters based on expression data of maize BR-related genes in meristem region of ninth internode. Clusters showing similar or opposite patterns as target trait are marked with red rectangles. (**b**) Heatmap of all clusters in meristem region of ninth internode. (**c**) Expression levels of genes in cluster 2 at meristem region of ninth internode. (**d**) Expression levels of genes in cluster 4 at meristem region of ninth internode. (**e**) Expression levels of genes in cluster 5 at meristem region of ninth internode. (**f**) Clusters based on expression data of maize BR-related genes in elongation region of ninth internode. Clusters showing similar or opposite patterns as target trait are marked with red rectangles. (**g**) Heatmap of all clusters in elongation region of ninth internode. (**h**) Expression levels of genes in cluster 2 at elongation region of ninth internode. (**i**) Expression levels of genes in cluster 6 at elongation region of ninth internode. (**j**) Expression pattern analysis of maize BR-related genes in response to coronatine. (**k**) Gene co-expression network analysis of maize BR-related genes. Genes with high expression are marked with a red asterisk.

**Figure 5 ijms-23-05551-f005:**
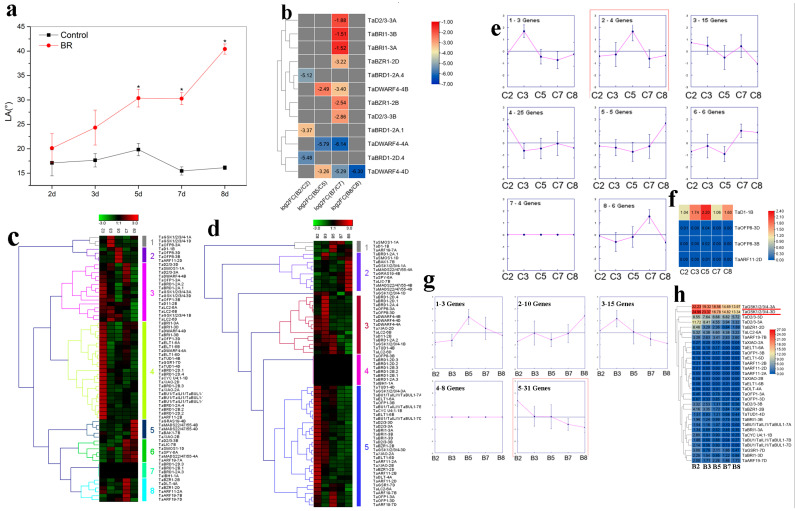
Trends and expression pattern analysis of wheat BR-related genes in response to BR. (**a**) Effect of BR on leaf angle in wheat seedlings. (**b**) Differentially expressed wheat BR-related genes between BR (**b**) and control (**c**) groups. FC: fold-change. (**c**) Heatmap of all clusters in control group. (**d**) Heatmap of all clusters in BR group. (**e**) Clusters based on expression data of wheat BR-related genes in control group. Cluster showing a similar pattern with target trait is marked with a red rectangle. (**f**) Expression levels of genes in cluster 2 in control group. (**g**) Clusters based on expression data of wheat BR-related genes in BR group. Cluster showing opposite pattern to target trait is marked with a red rectangle. (**h**) Expression levels of genes in cluster 5 in BR group. Genes with high expression levels are underlined in red.

**Figure 6 ijms-23-05551-f006:**
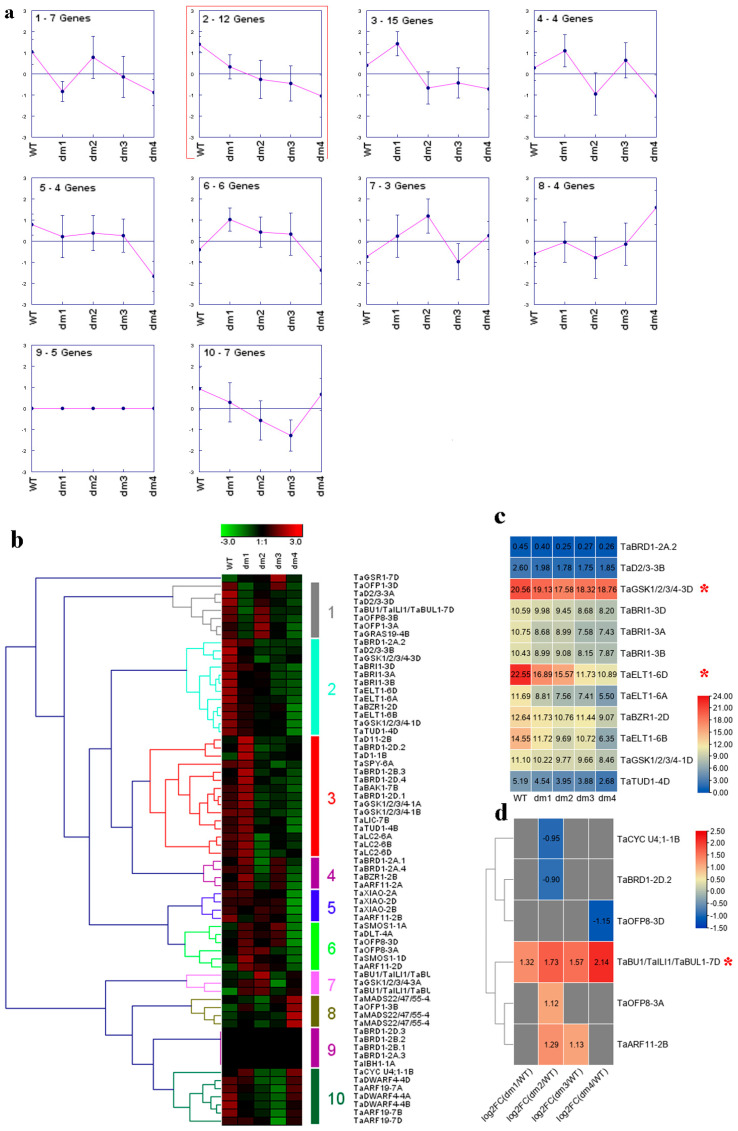
Trend analysis of wheat BR-related genes in wild-type (WT) and dwarf wheat. (**a**) Clusters based on expression data of wheat BR-related genes in WT, dm1, dm2, dm3, and dm4. Cluster showing a similar pattern to target trait is marked with a red rectangle. (**b**) Heatmap of all clusters. (**c**) Expression levels of genes in cluster 2. Genes with high expression are marked with red asterisks. (**d**) Differentially expressed wheat BR-related genes between dm and WT wheat seedlings. LogFC data represent up-regulation or down-regulation. FC: fold-change.

**Figure 7 ijms-23-05551-f007:**
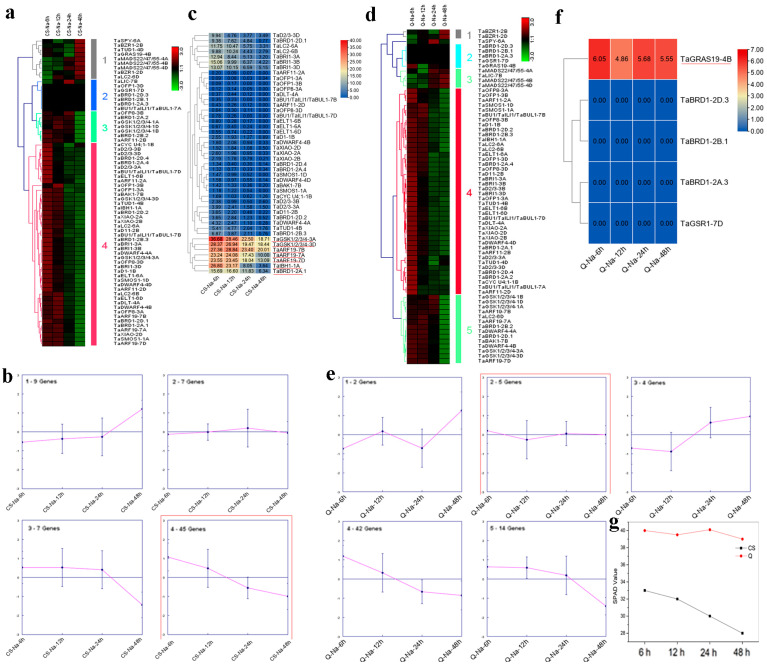
Trend analysis of wheat BR-related genes in response to salt. (**a**) Heatmap of all clusters in CS. (**b**) Clusters based on expression data of wheat BR-related genes in CS. Cluster showing a similar pattern with target trait is marked with a red rectangle. (**c**) Expression levels of genes in cluster 4 in CS. Genes with high expression are marked with red underlines. (**d**) Heatmap of all clusters in QM. (**e**) Clusters based on expression data of wheat BR-related genes in QM. Cluster showing a similar pattern with target trait is marked with a red rectangle. (**f**) Expression levels of genes in cluster 2. Genes with high expression are underlined in red. (**g**) Effect of salt on chlorophyll content in CS and QM.

**Figure 8 ijms-23-05551-f008:**
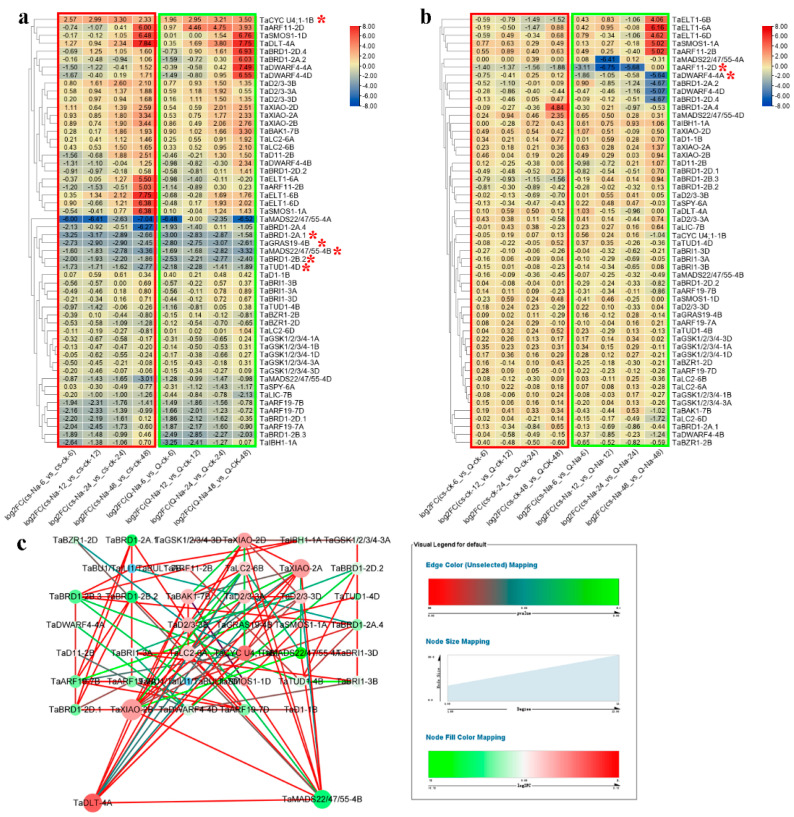
Expression analysis of wheat BR-related genes in response to salt. (**a**) DEG analysis between control and salt-treated groups in CS and QM. DEGs in both CS and QM at all time points are marked with red asterisks. (**b**) DEG analysis between CS and QM. DEGs in both CS and QM at all time points are marked with red asterisks. (**c**) Gene co-expression network analysis of wheat BR-related genes.

**Figure 9 ijms-23-05551-f009:**
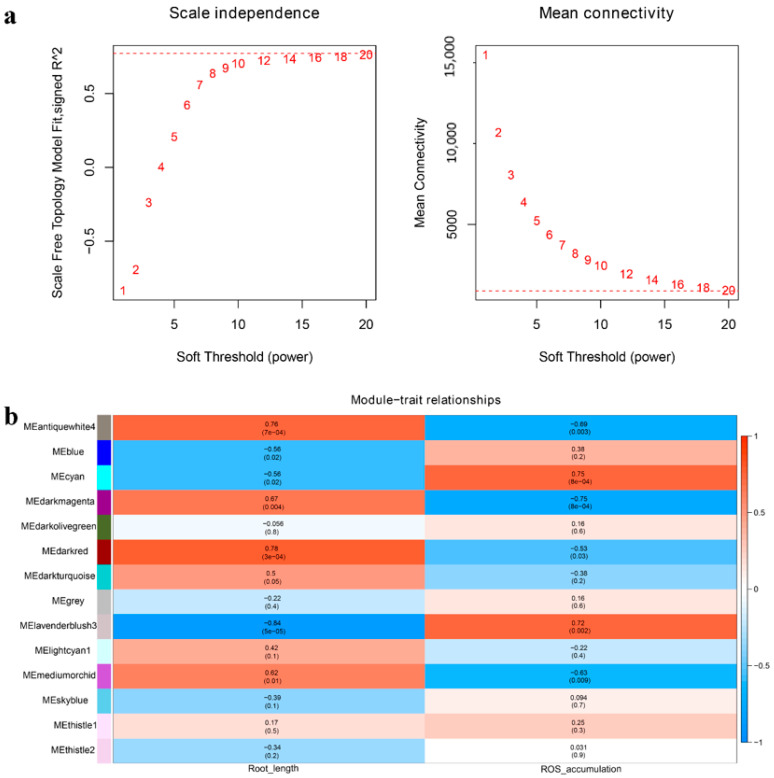
WGCNA of wheat genes identified in CS and QM in response to salt stress. (**a**) Scale independence and mean connectivity curves. Horizontal axis on left and right figures indicates power value, and vertical axis represents a fitting degree of adjacency matrix to a scale-free network. Vertical axis on right figure represents average connectivity of each node in the network. (**b**) Module-trait relationships showing significance of module eigengene correlation with traits (root length and ROS accumulation). Left panel shows 14 modules. Module-trait correlation ranging from −1 (blue) to 1 (red) is indicated with color scale on right.

**Figure 10 ijms-23-05551-f010:**
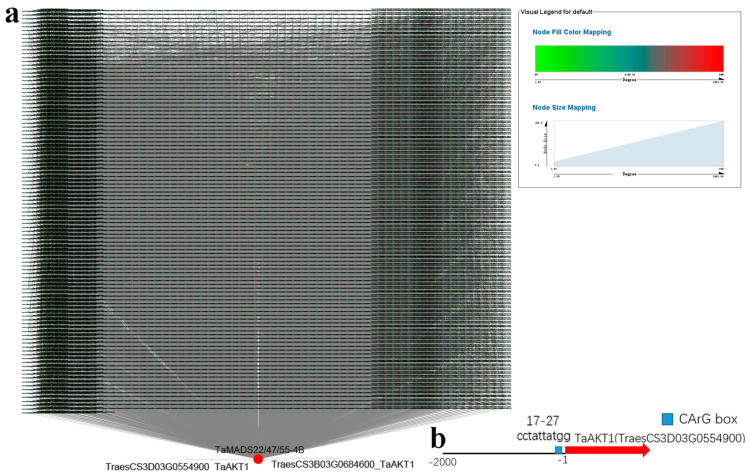
Hub genes identified in ‘darkred’ module. (**a**) Cytoscape representation of relationship of genes (edge weight ≥ 0.10) in ‘darkred’ module, genes with a large and small degree are indicated with large red and small green circles. (**b**) MADS binding *cis*-element (CArG) box in promoter of *TaAKT1*.

## Data Availability

The datasets used in this study were deposited in the NCBI Gene Expression Omnibus (15 May 2022, http://www.ncbi.nlm.nig.gov/geo) with accession number GSE115796, NCBI Sequence Read Archive (SRA) (15 May 2022, http://www.ncbi.nlm.nih.gov/sra) with accession number PRJNA633707, NCBI under BioProject ID PRJNA417210 with SRA submission ID SUB3198378, and NCBI SRA database under SRP062745. The raw RNA-seq data used to identify BR-mediated LA were deposited in the NCBI SRA database under accession number SRP157960. The datasets used and/or analyzed in the current study are available from the corresponding author upon reasonable request.

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
