# Peer review of "Genome-Wide Identification of Gramineae Brassinosteroid-Related Genes and Their Roles in Plant Architecture and Salt Stress Adaptation"

_ijms, 2022, doi:10.3390/ijms23105551_

Round 1

Reviewer 1 Report

For the purpose of elucidating the function of specific genes, it would be more research beneficial to focus on one particular aspect, or at least related aspects. From this perspective, I do not consider the combination of plant architecture and salt stress adaptation to be appropriate in terms of functional gene analysis. Moreover, also methodologically, there is a complete lack of data on the applied abiotic stress factor and the methodological procedure.

The authors state that they investigated expression profiles of several brassinosteroid-related genes in wheat and maize involved in salt stress adaptation. Subchapter of 4.3. describes only one wheat genotype analysis, which seedlings were treated by 24-epitestosterone of one concentration. Is testing the impact of one concentration variant sufficient? This part requires considerable refinement.

Genome-wide analyses were performed on four Gramineae species. What reasons led the authors to select only two species for transcriptomic analyses, while the methodology itself describes only one species (wheat)?

All figures in Supplementary folder do not contain the necessary information about the applied software used for their construction and also lack descriptive data for individual figures.

I am sorry, but I think that the manuscript would have sufficient scientific value as long as it deals with the problem from the perspective of plant architecture and experimental transcriptomic analyses are added.

Reviewer 2 Report

In the paper by Huang et al., the authors aimed at identifying genes regulated by brassinosteroids, which influence morphological traits such as plant height and leaf angle, and physiological traits such as salt stress tolerance. Starting from rice genes, the orthologues were identified in bread wheat, barley, maize and Sorghum bicolor genome, and the genes and corresponding proteins were characterized in detail. A study of the expression of these genes was also performed in different samples, timepoints and under different treatments.

As stated in the introduction of the paper, brassinosteroids are a class of plant growth regulators which are gaining increasing importance; therefore, a study providing new insights on this topic would be very useful.

That said, this paper suffers from a series of flaws, which require a serious and accurate revision of the text structure and included figures and tables. The analyses conducted yielded a huge amount if data and information, which are presented in form of text, tables and figures. The first part presents the BR-regulated genes identified in the four crop species starting from the corresponding rice proteins, and an enormous panel of analyses and data is shown. The second part, dealing with the expression analysis of BR-regulated genes, is in my opinion too dense to be of use for a reader; it provides too much information and a final take-home message is very difficult to deduce. The figures are too complicated, the letters within are too small, and it took a while, for me, to find out how to interpret plots and heatmaps. All the parts should be more reader-friendly.

What is perhaps the most serious defect of the paper, is that the Results section is far too verbose, made of over nine pages of text, 10 figures, 10 supplementary figures and 10 supplementary tables. However, the Discussion section is only less than one-page-and-a-half long. The two parts are far too unbalanced; it's hard for me to believe that all the results shown have been adequately commented and investigated in such a short discussion; probably the Results should be shortened and only the essential information should be included and then treated in the Discussion; alternatively the whole work should be splitted into two separate papers. This would also value the outstanding amount of work performed. Also in this case, the authors should ensure that figures are comprehensible and easy-to-understand.

Therefore, I regret that I cannot consider this paper publishable in the present form; the authors should accurately address the suggestions and criticisms in it, and submit a new and deeply-revised version.

I listed below some more focused comments, on a line-by-line basis:

L38: "including LA and PH" has no sense here

L52-55: check the brackets

L65-74: check the English; moreover, L66-68: I cannot understand the use of "import" and "these crop architecture" in this context

L85-87: unlike what is stated in the text, slight differences in bar thickness are visible between species; maybe a table would be better

L117-143: all this part must be improved; the references to Fig. S2-3 are actually to Fig. S2-8; at L143: the reference is to group C, not B

L150: what do you mean by "consistent"?

L152: I cannot understand the reference to fig. S5 in this context

L154: remove the reference to fig. S5

L171-172: please clarify

L306: "wildly" should be "widely"

L314-315: the verb is missing

L324: is the reference to tab. S10 correct here?

Figures:

Fig. 3-7: in my opinion the plots are not designed in a proper way. Each one shows three different samples (low-, medium- and high-sized), and three developmental stages for each, a single dashed line is shown as if it was referred to a whole timelapse of a unique sample. The results would be much clearer if each cluster were represented by 3 stacked plots, with 3 time points.

Fig. S3: in the caption, it would be useful to add: "compared to rice orthologues".

Author Response

Comments and Suggestions for Authors

In the paper by Huang et al., the authors aimed at identifying genes regulated by brassinosteroids, which influence morphological traits such as plant height and leaf angle, and physiological traits such as salt stress tolerance. Starting from rice genes, the orthologues were identified in bread wheat, barley, maize and Sorghum bicolor genome, and the genes and corresponding proteins were characterized in detail. A study of the expression of these genes was also performed in different samples, timepoints and under different treatments.

As stated in the introduction of the paper, brassinosteroids are a class of plant growth regulators which are gaining increasing importance; therefore, a study providing new insights on this topic would be very useful.

That said, this paper suffers from a series of flaws, which require a serious and accurate revision of the text structure and included figures and tables. The analyses conducted yielded a huge amount if data and information, which are presented in form of text, tables and figures. The first part presents the BR-regulated genes identified in the four crop species starting from the corresponding rice proteins, and an enormous panel of analyses and data is shown. The second part, dealing with the expression analysis of BR-regulated genes, is in my opinion too dense to be of use for a reader; it provides too much information and a final take-home message is very difficult to deduce. The figures are too complicated, the letters within are too small, and it took a while, for me, to find out how to interpret plots and heatmaps. All the parts should be more reader-friendly.

Thank you for your suggestion.

The Results have been shortened and the essential information were included and then treated in the Discussion. To make manuscript more reader-friendly, the letters in Figures 4, 5 and 7 were enlarged, and the clusters in Figure 3 and 4 have been represented by 3 stacked plots with 3 time points. Moreover, the manuscript has been revised and edited by a native English speaking colleague.

What is perhaps the most serious defect of the paper, is that the Results section is far too verbose, made of over nine pages of text, 10 figures, 10 supplementary figures and 10 supplementary tables. However, the Discussion section is only less than one-page-and-a-half long. The two parts are far too unbalanced; it's hard for me to believe that all the results shown have been adequately commented and investigated in such a short discussion; probably the Results should be shortened and only the essential information should be included and then treated in the Discussion; alternatively the whole work should be splitted into two separate papers. This would also value the outstanding amount of work performed. Also in this case, the authors should ensure that figures are comprehensible and easy-to-understand.

Thank you for your suggestion.

The Results have been shortened and the essential information were included and then treated in the Discussion. The letters in Figures 4, 5 and 7 were enlarged, and the clusters in Figure 3 and 4 have been represented by 3 stacked plots with 3 time points to make them easy-to-understand.

Please see that in the result section of revised manuscript.

Therefore, I regret that I cannot consider this paper publishable in the present form; the authors should accurately address the suggestions and criticisms in it, and submit a new and deeply-revised version.

I listed below some more focused comments, on a line-by-line basis:

L38: "including LA and PH" has no sense here

Thank you for your suggestion. The mistake has been mixed.

L52-55: check the brackets

Thank you for your suggestion. The mistake has been mixed.

L65-74: check the English; moreover, L66-68: I cannot understand the use of "import" and "these crop architecture" in this context

Thank you for your suggestion. The mistake has been mixed. These results have been removed.

L85-87: unlike what is stated in the text, slight differences in bar thickness are visible between species; maybe a table would be better

Thank you for your suggestion. These results have been removed.

L117-143: all this part must be improved; the references to Fig. S2-3 are actually to Fig. S2-8; at L143: the reference is to group C, not B

Thank you for your suggestion. The mistakes have been mixed.

L150: what do you mean by "consistent"?

Thank you for your suggestion. The mistake has been mixed. These results have been removed.

L152: I cannot understand the reference to fig. S5 in this context

Thank you for your suggestion. The mistake has been mixed. Fig. S5 has been changed into Fig. S5-1.

L154: remove the reference to fig. S5

Thank you for your suggestion. The mistake has been mixed. Fig. S5 has been removed.

L171-172: please clarify

Thank you for your suggestion. These results have been removed.

L306: "wildly" should be "widely"

Thank you for your suggestion. The mistake has been mixed.

L314-315: the verb is missing

Thank you for your suggestion. The mistake has been mixed. The ‘interacting’ was changed into ‘interactions’.

L324: is the reference to tab. S10 correct here?

Thank you for your suggestion. The mistake has been mixed. Table S10 has been bracketed.

Figures:

Fig. 3-7: in my opinion the plots are not designed in a proper way. Each one shows three different samples (low-, medium- and high-sized), and three developmental stages for each, a single dashed line is shown as if it was referred to a whole timelapse of a unique sample. The results would be much clearer if each cluster were represented by 3 stacked plots, with 3 time points.

Thank you for your suggestion. The clusters in Figure 3 and 4 have been represented by 3 stacked plots, with 3 time points. And Fig. 5-7 were already designed as you are intended.

Fig. S3: in the caption, it would be useful to add: "compared to rice orthologues".

Thank you for your suggestion. The mistakes have been mixed.

Reviewer 3 Report

The work described in the manuscript is interesting, the results are clear, and the conclusions are sound and based on the results obtained. I think these results add new information in this research area.

Author Response

The work described in the manuscript is interesting, the results are clear, and the conclusions are sound and based on the results obtained. I think these results add new information in this research area.

Thank you!

Round 2

Reviewer 1 Report

Dear authors,

Thank you for considering and incorporating the manuscript editing suggestions. 

I wish you much success in your further research.